# ENVIRONMENT PREDICTIVE CODING
# FOR EMBODIED AGENTS

## ABSTRACT

We introduce *environment predictive coding*, a self-supervised approach to learn environment-level representations for embodied agents. In contrast to prior work on self-supervised learning for images, we aim to jointly encode a series of images gathered by an agent as it moves about in 3D environments. We learn these representations via a *zone prediction task*, where we intelligently mask out portions of an agent's trajectory and predict them from the unmasked portions, conditioned on the agent's camera poses. By learning such representations on a collection of videos, we demonstrate successful transfer to multiple downstream navigation-oriented tasks. Our experiments on the photorealistic 3D environments of Gibson and Matterport3D show that our method outperforms the state-of-the-art on challenging tasks with only a limited budget of experience.

## 1 INTRODUCTION

In visual navigation tasks, an intelligent embodied agent must move around a 3D environment using its stream of egocentric observations to sense objects and obstacles, typically without the benefit of a pre-computed map. Significant recent progress on this problem can be attributed to the availability of large-scale visually rich 3D datasets (Chang et al., 2017; Xia et al., 2018; Straub et al., 2019), developments in high-quality 3D simulators (Anderson et al., 2018b; Kolve et al., 2017; Savva et al., 2019a; Xia et al., 2020), and research on deep memory-based architectures that combine geometry and semantics for learning representations of the 3D world (Gupta et al., 2017; Henriques & Vedaldi, 2018; Chen et al., 2019; Fang et al., 2019; Chaplot et al., 2020b;c).

Deep reinforcement learning approaches to visual navigation often suffer from sample inefficiency, overfitting, and instability in training. Recent contributions work towards overcoming these limitations for various navigation and planning tasks. The key ingredients are learning good image-level representations (Das et al., 2018; Gordon et al., 2019; Lin et al., 2019; Sax et al., 2020), and using modular architectures that combine high-level reasoning, planning, and low-level navigation (Gupta et al., 2017; Chaplot et al., 2020b; Gan et al., 2019; Ramakrishnan et al., 2020a).

Prior work uses supervised image annotations (Mirowski et al., 2016; Das et al., 2018; Sax et al., 2020) and self-supervision (Gordon et al., 2019; Lin et al., 2019) to learn good image representations that are transferrable and improve sample efficiency for embodied tasks. While promising, such learned image representations only encode the scene in the nearby locality. However, embodied agents also need higher-level semantic and geometric representations of their history of observations, grounded in 3D space, in order to reason about the larger environment around them.

Therefore, a key question remains: *how should an agent moving through a visually rich 3D environment encode its series of egocentric observations?* Prior navigation methods build *environment-level* representations of observation sequences via memory models such as recurrent neural networks (Wijmans et al., 2020), maps (Henriques & Vedaldi, 2018; Chen et al., 2019; Chaplot et al., 2020b), episodic memory (Fang et al., 2019), and topological graphs (Savinov et al., 2018; Chaplot et al., 2020c). However, these approaches typically use hand-coded representations such as occupancy maps (Chen et al., 2019; Chaplot et al., 2020b; Ramakrishnan et al., 2020a; Karkus et al., 2019; Gan et al., 2019) and semantic labels (Narasimhan et al., 2020; Chaplot et al., 2020a), or specialize them by learning end-to-end for solving a specific task (Wijmans et al., 2020; Henriques & Vedaldi, 2018; Parisotto & Salakhutdinov, 2018; Cheng et al., 2018; Fang et al., 2019).

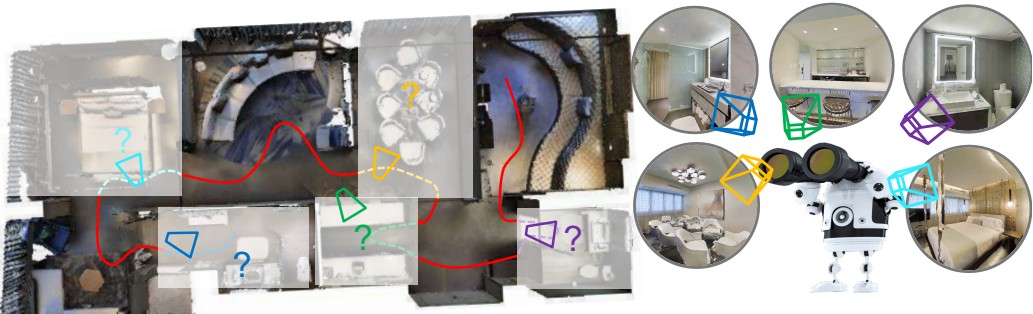

Figure 1: **Environment Predictive Coding:** During self-supervised learning, our model is given video walk-throughs of various 3D environments. We mask portions out of the trajectory (dotted lines) and learn to infer them from the unmasked parts (in red). We specifically mask out all overlapping views in a local neighborhood to limit the content shared with the unmasked views. The resulting EPC encoder builds environment-level representations of the seen content that are predictive of the unseen content (marked with a "?"), conditioned on the camera poses. The agent then uses this learned encoder in multiple navigational tasks in novel environments.

In this work, we introduce *environment predictive coding* (EPC), a self-supervised approach to learn flexible representations of 3D environments that are transferrable to a variety of navigation-oriented tasks. The key idea is to learn to encode a series of egocentric observations in a 3D environment so as to be predictive of visual content that the agent has not yet observed. For example, consider an agent that just entered the living room in an unfamiliar house and is searching for a refrigerator. It must be able to predict where the kitchen is and reason that it is likely to contain a refrigerator. The proposed EPC model aims to learn representations that capture these natural statistics of real-world environments in a self-supervised fashion, by watching videos recorded by other agents. See Fig. 1.

To this end, we devise a self-supervised *zone prediction* task in which the model learns environment embeddings by watching egocentric view sequences from other agents navigating in 3D environments in pre-collected videos. Specifically, we segment each video into zones of visually and geometrically connected views, while ensuring limited overlap across zones in the same video. Then, we randomly mask out zones, and predict the masked views conditioned on both the unmasked zones' views and the masked zones' camera poses. Intuitively, to perform this task successfully, the model needs to reason about the geometry and semantics of the environment to figure out what is missing. We devise a transformer-based model to infer the masked visual features. Our general strategy can be viewed as a context prediction task in sequential data (Devlin et al., 2018; Sun et al., 2019b; Han et al., 2019)—but, very differently, aimed at representing high-level semantic and geometric priors in 3D environments to aid embodied agents who act in them.

Through extensive experiments on Gibson and Matterport3D, we show that our method achieves good improvements on multiple navigation-oriented tasks compared to state-of-the-art models and baselines that learn image-level embeddings.

## 2 RELATED WORK

**Self-supervised visual representation learning**: Prior work leverages self-supervision to learn image and video representations from large collections of unlabelled data. Image representations attempt proxy tasks such as inpainting (Pathak et al., 2016) and instance discrimination (Oord et al., 2018; Chen et al., 2020; He et al., 2020), while video representation learning leverages signals such as temporal consistency (Wei et al., 2018; Fernando et al., 2017; Kim et al., 2019) and contrastive predictions (Han et al., 2019; Sun et al., 2019a). The VideoBERT project (Sun et al., 2019a;b) jointly learns video and text representations from unannotated videos via filling in masked out information. Dense Predictive Coding (Han et al., 2019; 2020) learns video representations that capture the slow-moving semantics in videos. Whereas these methods focus on capturing human activity for video recognition, we aim to learn geometric and semantic cues in 3D spaces for embodied agents. Accordingly, unlike the existing video models (Sun et al., 2019a;b; Han et al., 2019), our approach is grounded in the 3D relationships between views.

**Representation learning via auxiliary tasks for RL**: Reinforcement learning approaches often suffer from high sample complexity, sparse rewards, and unstable training. Prior work tackles these

challenges by using auxiliary tasks for learning image representations (Mirowski et al., 2016; Gordon et al., 2019; Lin et al., 2019; Shen et al., 2019; Ye et al., 2020). In contrast, we encode image sequences from embodied agents to obtain environment-level representations. Recent work also learns state representations via future prediction and implicit models (Ha & Schmidhuber, 2018; Eslami et al., 2018; Gregor et al., 2019; Hafner et al., 2019; Guo et al., 2020). In particular, neural rendering approaches achieve impressive reconstructions for arbitrary viewpoints (Eslami et al., 2018; Kumar et al., 2018). However, unlike our idea, they focus on pixelwise reconstruction, and their success has been limited to synthetically generated environments like DeepMind Lab (Beattie et al., 2016). In contrast to any of the above, we use egocentric videos to learn predictive feature encodings of photorealistic 3D environments to capture their naturally occurring regularities.

**Scene completion**: Past work in scene completion performs pixelwise reconstruction of 360 panoramas (Jayaraman & Grauman, 2018; Ramakrishnan et al., 2019), image inpainting (Pathak et al., 2016), voxelwise reconstructions of 3D structures and semantics (Song et al., 2017), and image-level extrapolation of depth and semantics (Song et al., 2018; Yang et al., 2019b). Recent work on visual navigation extrapolates maps of room-types (Wu et al., 2019; Narasimhan et al., 2020) and occupancy (Ramakrishnan et al., 2020a). While our approach is also motivated by anticipating unseen elements, we learn to extrapolate in a high-dimensional feature space (rather than pixels, voxels, or semantic categories) and in a self-supervised manner without relying on human annotations. Further, the proposed model learns from egocentric video sequences captured by other agents, without assuming access to detailed scans of the full 3D environment as in past work.

**Learning image representations for navigation**: Prior work exploits ImageNet pretraining (Gupta et al., 2017; Anderson et al., 2018a; Chen et al., 2019), mined object relations (Yang et al., 2019a), video (Chang et al., 2020), and annotated datasets from various image tasks (Sax et al., 2020; Chaplot et al., 2020c) to aid navigation. While these methods also consider representation learning in the context of navigation tasks, they are limited to learning image-level functions for classification and proximity prediction. In contrast, we learn predictive representations for sequences of observations conditioned on the camera poses.

## 3 APPROACH

We propose *environment predictive coding* (EPC) to learn self-supervised environment-level representations (Sec. 3.1). To demonstrate the utility of these representations, we integrate them into a transformer-based navigation architecture and refine them for individual tasks (Sec. 3.2). As we will show in Sec. 4, our approach leads to both better performance and better sample efficiency compared to existing approaches.

### 3.1 ENVIRONMENT PREDICTIVE CODING

Our hypothesis is that it is valuable for an embodied agent to learn a predictive coding of the environment. The agent must not just encode the individual views it observes, but also learn to leverage the encoded information to anticipate the unseen parts of the environment. Our key idea is that the environment embedding must be predictive of unobserved content, conditioned on the agent's camera pose. This equips an agent with the natural priors of 3D environments to quickly perform new tasks, like finding the refrigerator or covering more area.

We propose the proxy task of zone prediction to achieve this goal (see Fig. 2). For this task, we use a dataset of egocentric video walkthroughs collected parallely from other agents deployed in various unseen environments (Fig. 2, top). For each video, we assume access to RGB-D, egomotion data, and camera intrinsics. Specifically, our current implementation uses egocentric camera trajectories from photorealistic scanned indoor environments (Gibson (Xia et al., 2018)) to sample the training videos; we leave leveraging in-the-wild consumer video as a challenge for future work.

We do *not* assume that the agents who generated those training videos were acting to address a particular navigation task. In particular, their behavior need not be tied to the downstream navigation-oriented tasks for which we test our learned representation. For example, a training video may show agents moving about to maximize their area coverage, whereas the encoder we learn is applicable to an array of navigation tasks (as we will demonstrate in Sec. 4). Furthermore, we assume that the environments seen in the videos are *not* accessible for interactive training. In practice, this means

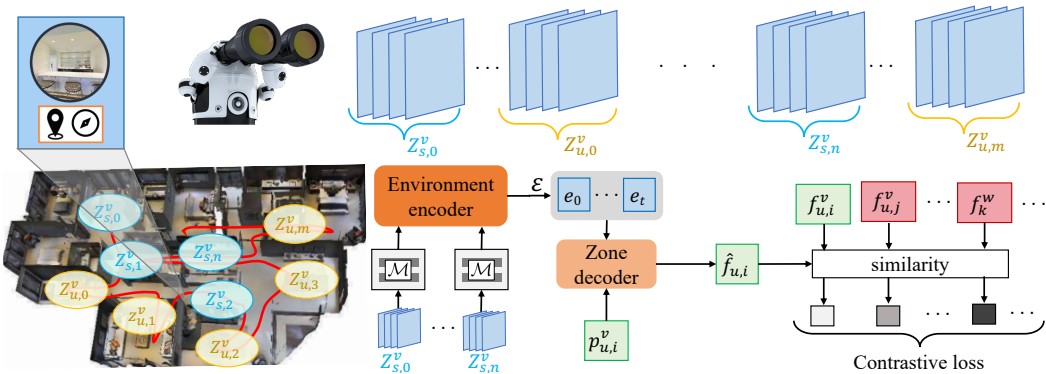

Figure 2: We propose the *zone prediction task* for self-supervised learning of environment embeddings from video walkthroughs generated by other agents. Each frame consists of the egocentric view and camera pose (top left). We group the frames in video $v$ into seen zones in cyan $\{Z_{s,0}^v, \cdots, Z_{s,n}^v\}$ and unseen zones in yellow $\{Z_{u,0}^v, \cdots, Z_{u,m}^v\}$ (top row). The zones are generated automatically based on viewpoint overlap in 3D space (bottom left). Given a camera pose $p_{u,i}^v$ sampled from the unseen zone $Z_{u,i}^v$, we use a transformer-based encoder-decoder architecture that generates environment embeddings $\mathcal{E}$ from the seen zones, and predicts the feature encoding $\hat{f}_{u,i}$ of $Z_{u,i}^v$ conditioned on the pose $p_{u,i}^v$ (bottom center). The model is trained to distinguish the positive $f_{u,i}^v$ from negatives in the same video $\{f_{u,j}^v\}_{j \neq i}$ as well from other videos $\{f_k^w\}_{t \neq s}$ (bottom right).

that we can parallelly collect data from different robots deployed in a large number of environments, without having to actually train our navigation policy on those environments. These assumptions are much weaker than those made by prior work on imitation learning and behavioral cloning that rely on *task-specific* data generated from experts (Bojarski et al., 2016; Giusti et al., 2016).

Our method works as follows. First, we automatically segment videos into "zones" which contain frames with significant view overlaps. We then perform the self-supervised zone prediction task on the segmented videos. Finally, we incorporate the learned environment encoder into an array of downstream navigation-oriented tasks. We explain each step in detail next.

**Zone generation** At a glance, one might first consider masking arbitrary individual frames in the training videos. However, doing so is inadequate for representation learning, since unmasked frames having high viewpoint overlap with the masked frame can make its prediction trivial. Instead, our approach masks *zones* of frames at once. We define a zone to be a set of frames in the video which share a significant overlap in their viewpoints. We also require that the frames across multiple zones share little to no overlap.

To generate these zones, we first cluster frames in the videos based on the amount of pairwise-geometric overlap between views. We estimate the viewpoint overlap $\psi(o_i, o_j)$ between two frames $o_i$, $o_j$ by measuring their intersection in 3D point clouds obtained by backprojecting depth inputs into 3D space. See Appendix for more details. For a video of length $L$, we generate a distance matrix $D \in \mathbb{R}^{L \times L}$ where $D_{i,j} = 1 - \psi(o_i, o_j)$. We then perform hierarchical agglomerative clustering (Lukasová, 1979) to cluster the video frames into zones based on $D$ (see Fig. 2, bottom left). While these zones naturally tend to overlap near their edges, they typically capture disjoint sets of content in the video. Note that the zones segment *video trajectories*, not floorplan maps, since we do not assume access to the full 3D environment.

**Zone prediction task** Having segmented the video into zones, we next present our EPC zone prediction task to learn environment embeddings (see Fig. 2). We randomly divide the video $v$ into seen zones $\{Z_{s,i}^v\}_{i=1}^n$ (cyan) and unseen zones $\{Z_{u,i}^v\}_{i=1}^m$ (yellow), where a zone $Z$ is a tuple of images and the corresponding camera poses $Z_i = \{(o_j, p_j)\}_1^{|Z_i|}$. Given the seen zones, and the camera pose from an unseen zone $p_{u,i}^v$, we need to infer a feature encoding of the unseen zone $Z_{u,i}^v$. To perform this task, we first extract visual features $x$ from each RGB-D frame $o$ in the video using pretrained CNNs (see Sec. 3.2). These features are concatenated with the corresponding pose $p$ and projected using an MLP $\mathcal{M}$ to obtain the image-level embedding. The target features for the unseen zone $Z_{u,i}^v$ are obtained as follows:

$$f_{u,i}^v = \frac{1}{|Z_{u,i}^v|} \sum_{[x,p] \in Z_{u,i}^v} \mathcal{M}([x,p]). \tag{1}$$

The rationale behind the feature averaging is that we want to predict the high-level visual content of the zone, while ignoring viewpoint specific variations within the zone.

We use a transformer-based encoder-decoder model to perform this task (Vaswani et al., 2017). Our model consists of an environment encoder and a zone decoder which infers the zone features (see Fig. 2, bottom). The environment encoder takes in the image-level embeddings $\mathcal{M}([x, p])$ from the input zones, and performs multi-headed self-attention to generate the environment embeddings $\mathcal{E}$. The zone decoder attends to $\mathcal{E}$ using the average camera pose from the unseen zone $p_{u,i}^v$ and predicts the zone features as follows:

$$\hat{f}_{u,i} = \text{ZoneDecoder}(\mathcal{E}, p_{u,i}^v). \tag{2}$$

We transform all poses in the input zones relative to $p_{u,i}^v$ before encoding, which provides the model an egocentric view of the world. The environment encoder, zone decoder, and the projection function $\mathcal{M}$ are jointly trained using noise-contrastive estimation (Gutmann & Hyvärinen, 2010). We use $\hat{f}_{u,i}$ as the anchor and $f_{u,i}^v$ from Eqn. 1 as the positive. We sample negatives from other unseen zones in the same video and from all zones in other videos. The loss for the $i^{\text{th}}$ unseen zone in video $v$ is:

$$L_i^v = -\log \frac{\exp\big(\text{sim}(\hat{f}_{u,i}, f_{u,i}^v)\big)}{\sum\limits_{j=1}^{m} \exp\big(\text{sim}(\hat{f}_{u,i}, f_{u,j}^v)\big) + \sum\limits_{w \neq v, k} \exp\big(\text{sim}(\hat{f}_{u,i}, f_k^w)\big)}, \tag{3}$$

where $\text{sim}(q, k) = \frac{q \cdot k}{|q||k|}\frac{1}{\tau}$ and $\tau$ is a temperature hyperparameter. The idea is to predict zone representations that are closer to the ground truth, while being sufficiently different from the negative zones. Since the unseen zones have only limited overlap with the seen zones, the model needs to effectively reason about the geometric and semantic context in the seen zones to differentiate the positive from the negatives. We discourage the model from simply capturing video-specific textures and patterns by sampling negatives from within the same video.

## 3.2 Environment embeddings for embodied agents

Having introduced our approach to learn environment embeddings in a self-supervised fashion, we now briefly overview how these embeddings are used for agents performing navigation-oriented tasks. To this end, we integrate our pre-trained environment encoder into the Scene Memory Transformer (SMT) (Fang et al., 2019). Our choice of SMT is motivated by the recent successes of transformers in both NLP (Devlin et al., 2018) and vision (Sun et al., 2019b; Fang et al., 2019). However, our idea is potentially applicable to other forms of memory models as well.

We briefly overview the SMT architecture (see Fig. 3, center). It consists of a scene memory that stores visual features $\{x_i\}_{i=0}^t$ and agent poses $\{p_i\}_{i=0}^t$ generated from the observations seen during an episode. The environment encoder uses self-attention on the history of observations to generate a richer set of environment embeddings $\{e_i\}_{i=1}^t$. At a given time-step $t + 1$, the policy decoder attends to the environment embeddings using the inputs $o_{t+1}$, which consist of the visual feature $x$ and agent pose $p$ at time $t + 1$. The outputs of the policy decoder are used to sample an action $a_{t+1}$ and estimate the value $v_{t+1}$. We detail each component in the Appendix.

To incorporate our EPC environment embeddings, we modify two key components from the original SMT model. First, and most importantly, we initialize the environment encoder with our pre-trained EPC (see Fig. 3, left). Second, we replace the end-to-end trained image encoders with MidLevel features that are known to be useful across a variety of embodied tasks (Sax et al., 2020) (see Fig. 3, right). We consider two visual modalities as inputs: RGB and depth. For RGB, we extract features from the pre-trained models in the max-coverage set proposed by Sax et al. (2020). These include surface normals, keypoints, semantic segmentation, and 2.5D segmentation. For depth, we extract features from pre-trained models that predict surface normals and keypoints from depth (Zamir et al., 2020). For training the model on a navigation task, we keep the visual features frozen, and only finetune the environment encoder, policy decoder, policy $\pi$, and value function $V$.

## 4 Experiments

We validate our pre-trained EPC environment embeddings for zone prediction (Sec. 4.1) and multiple downstream tasks that require an embodied agent to move intelligently through an unmapped

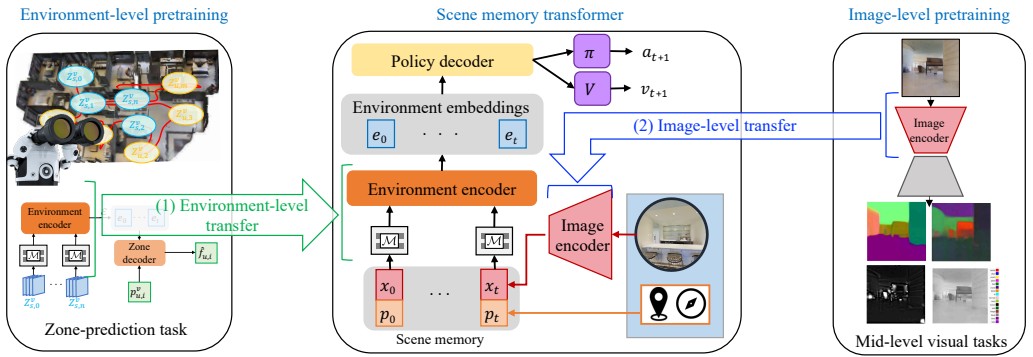

Figure 3: **Integrating environment-level pre-training for navigation: Left:** The first level of transfer occurs for the environment-level representations. We transfer the proposed EPC environment encoder and projection function $\mathcal{M}$ that are pre-trained for zone prediction. **Right:** The second level of transfer occurs for the image-level representations. We transfer a pre-trained MidLevel image encoder (Sax et al., 2020) to generate visual features for each input in the scene memory. **Center:** To train the SMT on a task, we keep the visual features frozen, and finetune the environment encoder and projection function $\mathcal{M}$ with the rest of the SMT model.

environment (Sec. 4.2). We evaluate the sensitivity of self-supervised learning to noise in the video data (Sec. 4.3), and assess noise robustness of the learned policies on downstream tasks (Sec. 4.4).

**EXPERIMENTAL SETUP AND TASKS**    We perform experiments on the Habitat simulator (Savva et al., 2019b) with Matterport3D (MP3D) (Chang et al., 2017) and Gibson (Xia et al., 2018), two challenging and photorealistic 3D datasets with $\sim 90$ and $500$ scanned real-world indoor environments, respectively. Our observation space consists of $171 \times 128$ RGB-D observations and odometry sensor readings that provide the relative agent pose $p = (x, y, \theta)$ w.r.t the agent pose at $t = 0$. Our action space consists of: MOVE-FORWARD by $25$cm, TURN-LEFT by $30°$, and TURN-RIGHT by $30°$. For all methods, we assume noise-free actuation and odometry for simplicity.

We use MP3D for interactive RL training, and reserve Gibson for evaluation. We use the default train/val/test split for MP3D (Savva et al., 2019b) for 1000-step episodes. For Gibson, which has smaller environments, we evaluate on the $14$ validation environments for 500-step episodes. Following prior work (Ramakrishnan et al., 2020a; Chaplot et al., 2020b), we divide results on Gibson into small and large environments. We generate walkthroughs for self-supervised learning from 332 Gibson training environments. We train a SMT(scratch) agent to perform area-coverage on MP3D. It explores starting from multiple locations and gathers the RGB-D and odometer readings for 500 steps per video. Note that this agent only collects data, and is not used for downstream tasks. This results in $\sim 5000$ videos, which we divide into an 80-20 train/val split.

We evaluate our approach on three standard tasks from the literature:

1. **Area coverage** (Chen et al., 2019; Chaplot et al., 2020b; Ramakrishnan et al., 2020b): The agent is rewarded for maximizing the area covered (in $\mathrm{m}^2$) within a fixed time budget.

2. **Flee** (Gordon et al., 2019): The agent is rewarded for maximizing the flee distance (in m), i.e., the geodesic distance between its starting location and the terminal location, for fixed-length episodes.

3. **Object coverage** (Fang et al., 2019; Ramakrishnan et al., 2020b): The agent is rewarded for maximizing the number of categories of objects covered during exploration (see Appendix). Since Gibson lacks extensive object annotations, we evaluate this task only on MP3D.

Together, these tasks capture different forms of geometric and semantic inference in 3D environments (e.g., area/object coverage encourage finding large open spaces/new objects, respectively).

**BASELINES**    We compare to the following baselines:

**Scratch baselines**: We randomly initialize the visual encoders and policy and train them end-to-end for each task. Images are encoded using ResNet-18. Agent pose and past actions are encoded using FC layers. These are concatenated to obtain the features at each time step. We use three temporal aggregation schemes. *Reactive (scratch)* has no memory. *RNN (scratch)* uses a 2-layer LSTM as the temporal memory. *SMT (scratch)* uses a Scene Memory Transformer for aggregating observations (Fang et al., 2019).

| Method | 4-2 split | | 2-4 split | |
| --- | --- | --- | --- | --- |
| | w/ inputs | w/o inputs | w/ inputs | w/o inputs |
| Nearest neighbors | 0.033 | 0.591 | 0.011 | 0.295 |
| Random masking | 0.332 | 0.760 | 0.209 | 0.449 |
| EPC (bilinear) | 0.452 | 0.806 | 0.321 | 0.518 |
| EPC ($\ell_2$-norm) | **0.528** | **0.825** | **0.368** | **0.556** |

Table 1: **Zone prediction:** Top-1 zone prediction accuracy on the validation walkthrough videos.

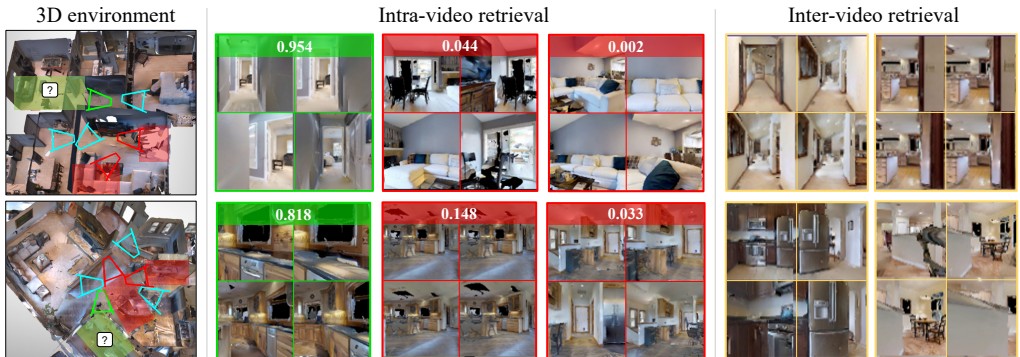

Figure 4: Each row shows one zone prediction example. **Left:** Top-down view of the 3D environment from which the video was sampled. The cyan viewing frusta correspond to the average pose for three input zones. Given the images and camera poses from each input zone, and a target camera pose (green frustum), the model predicts the corresponding zone feature (the masked green zone). **Center:** Given the inferred feature, we rank three masked (unobserved) zones from within the same video, where green is the positive zone and the red are the negatives. For each zone, we show four randomly sampled images along with the retrieval confidence. Our method retrieves the positive with high confidence. The model correctly predicts the existence of the narrow corridor (top row) and a kitchen counter (bottom row) given the target poses. **Right:** Top two retrieved zones from *other* videos that are closest to the inferred feature. The features predicted by the model are general enough to retrieve related concepts from other videos (narrow corridors and kitchens).

**SMT (MidLevel)**: extracts image features from pre-trained encoders that solve various mid-level perceptual tasks (Sax et al., 2020). This is an ablation of our model from Sec. 3.2 that uses the same image features, but randomly initializes the environment encoder. This SoTA image-level encoder is a critical baseline to show the impact of our proposed EPC environment-level encoder.

**SMT (Video)**: Inspired by Dense Predictive Coding (Han et al., 2019), this baseline uses MidLevel features and pre-trains the environment encoder as a video-level model using the same training videos as our model. For pre-training, we randomly sample 25 consecutive frames as inputs and predict the average features corresponding to the next 15 frames. We query based on the time (not pose) and train the model using the NCE loss in Eqn. 3.

**OccupancyMemory**: This is similar to the SoTA Active Neural SLAM model (Chaplot et al., 2020b) that maximizes area coverage, but using ground-truth depth to build the map (instead of RGB) and a state-of-the-art pointnav agent (Wijmans et al., 2020) for low-level navigation (instead of a planner). It represents the environment as a top-down occupancy map.

All models are trained in PyTorch (Paszke et al., 2019) with DD-PPO (Wijmans et al., 2020) for 15M frames with 64 parallel processes and the Adam optimizer. See Appendix.

## 4.1 ZONE PREDICTION PERFORMANCE

First we evaluate the EPC embedding quality in terms of zone prediction on the validation videos. We divide each video into $m$ seen and $n$ unseen zones and infer the features for each unseen zone, given its average camera pose. We rank the features from the $n$ unseen zones based on their similarity with the inferred feature, and measure the top-1 retrieval accuracy. We evaluate with $(m = 4, n = 2)$ and $(m = 2, n = 4)$ splits. The larger the value of $m$, the easier the task, since more information is available as input. We also test two simple baselines. **Nearest neighbors** uses the query pose to retrieve the 50 closest frames in the input zones, and outputs their averaged features. **Random**

| | Area coverage (m$^2$) | | | Flee (m) | | | Object coverage (#obj) | |
|---|---|---|---|---|---|---|---|---|
| Method | Gibson-S | Gibson-L | MP3D | Gibson-S | Gibson-L | MP3D | MP3D-cat. | MP3D-inst. |
| Reactive (scratch) | $17.4 \pm 0.2$ | $22.8 \pm 0.6$ | $68.0 \pm 1.3$ | $1.9 \pm 0.1$ | $2.5 \pm 0.3$ | $5.1 \pm 0.3$ | $6.2 \pm 0.0$ | $19.0 \pm 0.2$ |
| RNN (scratch) | $20.6 \pm 0.5$ | $28.6 \pm 0.3$ | $79.1 \pm 2.1$ | $2.3 \pm 0.2$ | $2.8 \pm 0.4$ | $5.9 \pm 0.1$ | $6.1 \pm 0.0$ | $18.6 \pm 0.2$ |
| SMT (scratch) | $23.0 \pm 0.7$ | $32.3 \pm 0.8$ | $104.8 \pm 2.3$ | $3.3 \pm 0.2$ | $4.4 \pm 0.4$ | $6.9 \pm 0.6$ | $7.0 \pm 0.1$ | $23.2 \pm 0.9$ |
| SMT (MidLevel) | $29.1 \pm 0.1$ | $47.2 \pm 1.7$ | $155.7 \pm 2.0$ | $4.2 \pm 0.0$ | $6.0 \pm 0.4$ | $10.6 \pm 0.3$ | $7.6 \pm 0.1$ | $26.8 \pm 0.6$ |
| SMT (Video) | $28.7 \pm 0.5$ | $50.6 \pm 2.6$ | $129.7 \pm 2.8$ | $4.1 \pm 0.0$ | $5.0 \pm 0.6$ | $10.9 \pm 0.5$ | $7.3 \pm 0.1$ | $25.4 \pm 1.0$ |
| OccupancyMemory | $29.4 \pm 0.1$ | $\mathbf{67.4} \pm 0.9$ | $155.6 \pm 1.4$ | $2.8 \pm 0.0$ | $7.0 \pm 0.4$ | $\mathbf{14.1} \pm 0.6$ | $7.8 \pm 0.1$ | $27.8 \pm 0.4$ |
| Ours (EPC) | $\mathbf{29.9} \pm 0.3$ | $56.4 \pm 2.1$ | $\mathbf{165.6} \pm 2.8$ | $\mathbf{4.5} \pm 0.1$ | $\mathbf{7.1} \pm 0.4$ | $12.8 \pm 0.6$ | $\mathbf{8.6} \pm 0.1$ | $\mathbf{34.5} \pm 0.8$ |

Table 2: **Downstream task performance** at the end of the episode. Gibson-S/L means small/large. MP3D-cat./inst. means categories/instances. All methods are evaluated on three random seeds. See Appendix for performance vs. time step plots.

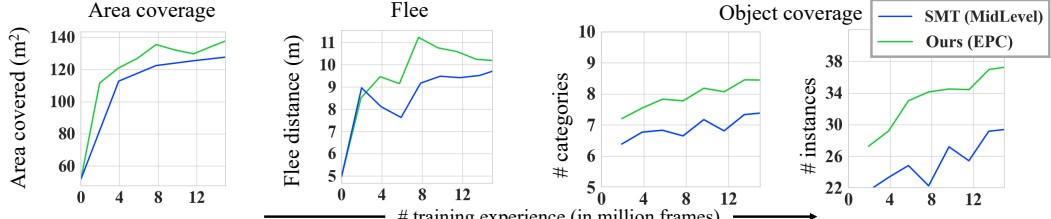

Figure 5: **Sample efficiency** on Matterport3D val split. Our environment-level pre-training leads to 2-4× training sample efficiency when compared to SoTA image-level pre-training. See Appendix for Gibson plots.

**masking** uses a different proxy task to learn the environment representations, randomly masking out 10 consecutive frames in the video and predicting their averaged feature from the rest. **EPC (bilinear)** uses bilinear product similarity (Oord et al., 2018) instead of the $\ell_2$-norm below Eqn. 3. We report retrieval from only the unseen zones (w/o inputs) as well as the more challenging case where input zones are also candidates (w/inputs).

Tab. 1 shows the results. The EPC ($\ell_2$-norm) model obtains superior retrieval performance on both settings. It retrieves the positive zones with high confidence (see Fig. 4 and Appendix). EPC's gain over random masking shows the value of the proposed zone generation step. Therefore, we select this model for downstream task transfer.

## 4.2 DOWNSTREAM TASK PERFORMANCE

Now we transfer these features to downstream navigation tasks. Tab. 2 shows the results. On both datasets, we observe the following ordering:

$$\text{Reactive (scratch)} \; < \; \text{RNN (scratch)} \; < \; \text{SMT (scratch)}. \tag{4}$$

This is in line with results reported by Fang et al. (2019) and verifies our implementation of SMT. Using MidLevel features for SMT leads to significant gains in performance versus training image encoders from scratch.

Our environment-level pre-training provides substantial improvements compared to SMT (MidLevel), particularly for larger environments. Furthermore, SMT (Video)—the video-level pre-training strategy—often deteriorates performance compared to using only image-level pre-training. This highlights EPC's value in representing the underlying 3D spaces of the walkthroughs instead of treating them simply as video frames. EPC competes closely and even slightly outperforms the state-of-the-art OccupancyMemory on these tasks, with a significant gain on the object coverage metrics. Thus, our model competes strongly with a *task-specific* representation model on the tasks that the latter was designed for, while outperforming it significantly on other tasks.

Finally, Fig. 5 shows that EPC offers better sample efficiency than image-only pre-training: our method reaches the best performance of SMT (MidLevel) 2-4× faster. This confirms our hypothesis: transferring environment-level representations learned via contextual reasoning can help embodied agents learn faster compared to the current approach of transferring image-level encoders alone.

| | Area coverage (m$^2$) | | | Flee (m) | | | Object coverage (#obj) | |
|---|---|---|---|---|---|---|---|---|
| Method | Gibson-S | Gibson-L | MP3D | Gibson-S | Gibson-L | MP3D | MP3D-cat. | MP3D-inst. |
| SMT (MidLevel) | $29.1 \pm 0.1$ | $47.2 \pm 1.7$ | $155.7 \pm 2.0$ | $4.2 \pm 0.0$ | $6.0 \pm 0.4$ | $10.6 \pm 0.3$ | $7.6 \pm 0.1$ | $26.8 \pm 0.6$ |
| EPC | $29.9 \pm 0.3$ | $56.4 \pm 2.1$ | $165.6 \pm 2.8$ | $4.5 \pm 0.1$ | $7.1 \pm 0.4$ | $12.8 \pm 0.6$ | $8.6 \pm 0.1$ | $34.5 \pm 0.8$ |
| EPC w/ noisy depth | $30.6 \pm 0.4$ | $58.0 \pm 3.8$ | $163.2 \pm 3.8$ | $4.8 \pm 0.0$ | $7.7 \pm 0.4$ | $13.0 \pm 0.5$ | $8.3 \pm 0.0$ | $31.8 \pm 0.0$ |
| EPC w/ noisy depth and pose | $31.2 \pm 0.3$ | $56.4 \pm 1.8$ | $149.6 \pm 0.4$ | $4.4 \pm 0.0$ | $7.7 \pm 0.2$ | $12.4 \pm 0.4$ | $7.8 \pm 0.1$ | $28.9 \pm 1.1$ |
| EPC w/ simple heuristic | $30.1 \pm 0.2$ | $59.9 \pm 1.6$ | $166.2 \pm 3.1$ | $4.6 \pm 0.1$ | $7.3 \pm 0.5$ | $12.3 \pm 0.4$ | $8.2 \pm 0.1$ | $32.5 \pm 0.2$ |

Table 3: Impact of noisy video data and a non-learned policy for video generation on EPC self-supervised learning.

| | Area coverage (m$^2$) | | | Flee (m) | | | Object cov. (#cat.) | | |
|---|---|---|---|---|---|---|---|---|---|
| Method | NF | N-D | N-D,P | NF | N-D | N-D,P | NF | N-D | N-D,P |
| SMT (MidLevel) | $155.7 \pm 2.0$ | $145.1 \pm 2.3$ | $134.2 \pm 1.8$ | $10.6 \pm 0.3$ | $10.6 \pm 0.6$ | $10.8 \pm 0.4$ | $7.6 \pm 0.2$ | $7.3 \pm 0.1$ | $7.3 \pm 0.2$ |
| OccupancyMemory | $155.6 \pm 1.4$ | $86.6 \pm 2.2$ | $85.2 \pm 2.4$ | $\mathbf{14.1} \pm 0.6$ | $10.9 \pm 0.2$ | $10.2 \pm 0.3$ | $7.8 \pm 0.1$ | $5.8 \pm 0.0$ | $5.8 \pm 0.0$ |
| EPC | $\mathbf{165.6} \pm 2.8$ | $\mathbf{148.4} \pm 1.4$ | $\mathbf{152.3} \pm 2.2$ | $12.8 \pm 0.6$ | $\mathbf{12.2} \pm 0.1$ | $\mathbf{11.4} \pm 0.2$ | $\mathbf{8.6} \pm 0.1$ | $\mathbf{8.4} \pm 0.2$ | $\mathbf{8.4} \pm 0.2$ |

Table 4: Comparing robustness to sensor noise on downstream tasks in Matterport3D. Note: NF denotes noise free sensing, N-D denotes noisy depth (and noise-free pose), and N-D,P denotes noisy depth and pose. See Appendix G for full results.

### 4.3 SENSITIVITY ANALYSIS OF SELF-SUPERVISED LEARNING

We analyze the sensitivity of EPC to sensory noise in the videos, and the exploration strategy used for video data collection. Specifically, we inject noise in the depth and pose data from the videos using existing noise models from Choi et al. (2015) and Ramakrishnan et al. (2020a). We also replace the video walkthroughs from the area-coverage agent with an equivalent amount of data collected by a simple heuristic used in prior work (Chen et al., 2019; Ramakrishnan et al., 2020b). It works as follows: move forward until colliding, then turn left or right by a random amount, then continue moving forward. We evaluate the impact of these changes on the downstream task performance.

See Tab. 3. Our approach EPC is reasonably robust to changes in the video data during SSL training. The performance remains stable when noise is injected into depth inputs. While it starts to decline on MP3D when we further inject noise into pose inputs, EPC still generally outperforms the random initialization of environment-encoder in SMT (MidLevel). Note that we do not employ any noise-correction mechanism, which could better limit this decline (Chaplot et al., 2020b; Ramakrishnan et al., 2020a). Finally, the performance is not significantly impacted when we use video data from a simple exploration heuristic, showing that EPC does not require a strong exploration policy for the agent that generates the self-supervised training videos, nor does it require a tight similarity between the tasks demonstrated in the videos and the downstream tasks.

### 4.4 ROBUSTNESS OF LEARNED POLICIES TO SENSOR NOISE

In previous experiments, we assumed the availability of ground-truth depth and pose sensors for downstream tasks. Now, we relax these assumptions and re-evaluate all methods by injecting noise in the depth and pose sensors for downstream tasks (same noise models as Sec. 4.3), without any noise-correction. This is a common evaluation protocol for assessing noise robustness (Chen et al., 2019; Ramakrishnan et al., 2020b). We compare the top three methods on MP3D in Tab. 4 and provide the complete set of results in Appendix G. As expected, the performance declines slightly as we add noise to more sensors (depth, then pose). However, most learned approaches are reasonably stable. EPC outperforms all methods when all noise sources are added. OccupancyMemory declines rapidly in the absence of noise-correction due to accumulated errors in the map.

## 5 CONCLUSIONS

We introduced Environment Predictive Coding, a self-supervised approach to learn environment-level representations for embodied agents. By training on video walkthroughs generated by other agents, our model learns to infer missing content through a zone-prediction task. When transferred to multiple downstream embodied agent tasks, the resulting embeddings lead to better performance and sample-efficiency compared to the current practice of transferring only image-level representations. In future work, we plan to extend our idea for goal-driven tasks like PointNav and ObjectNav.

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

# Appendix

## A    ZONE GENERATION

As discussed in the main paper, we generate zones by first clustering frames in the video based on their geometric overlap. Here, we provide details on how this overlap is estimated. First, we project pixels in the image to 3D point-clouds using the camera intrinsics and the agent pose. Let $D_i$, $p_i$ be the depth map and agent pose for frame $i$ in the video. The agent's pose in frame $i$ can be expressed as $p_i = (\boldsymbol{R}_i, \boldsymbol{t}_i)$, with $\boldsymbol{R}_i, \boldsymbol{t}_i$ representing the *agent's* camera rotation and translation in the world coordinates. Let $\boldsymbol{K} \in \mathbb{R}^{3\times3}$ be the intrinsic camera matrix, which is assumed to be provided for each video. We then project each pixel $x_{ij}$ in the depth map $D_i$ to the 3D point cloud as follows:

$$w_{ij} = \begin{bmatrix} \boldsymbol{R}_i & \boldsymbol{t}_i \\ \boldsymbol{0} & 1 \end{bmatrix} \boldsymbol{K}^{-1} x_{ij}, \ \forall j \in \{1, ..., S_i\} \tag{5}$$

where $S_i$ is the total number of pixels in $D_i$. By doing this operation for each pixel, we can obtain the point-cloud $\boldsymbol{W}_i$ corresponding to the depth map $D_i$. To compute the geometric overlap between two frames $i$ and $j$, we estimate the overlap in their point-clouds $\boldsymbol{W}_i$ and $\boldsymbol{W}_j$. Specifically, for each point $w_i \in \boldsymbol{W}_i$, we retrieve the nearest neighbor from $w_j \in \boldsymbol{W}_j$ and check whether the pairwise distance in 3D space is within a threshold $\tau$: $||w_i - w_j||_2 < \tau$ . If this condition is satisfied, then a match exists for $w_i$. Then, we define the overlap fraction $\psi(D_i, D_j)$ the fraction of points in $\boldsymbol{W}_i$ which have a match in $\boldsymbol{W}_j$. This overlap fraction is computed pairwise between all frames in the video, and hierarchical agglomerative clustering is performed using this similarity measure.

## B    TASK DETAILS

For the object coverage task, to determine if an object is covered, we check if it is within $3\text{m}$ of the agent, present in the agent's field of view, and if it is not occluded (Ramakrishnan et al., 2020b). We use a shaped reward function:

$$R_t = O_t - O_{t-1} + 0.02(C_t - C_{t-1}), \tag{6}$$

where $O_t$, $C_t$ are the number of object categories and 2D grid-cells visited by time $t$ (similar to Fang et al. (2019)).

## C    SCENE MEMORY TRANSFORMER

We provide more details about individual components of the Scene Memory Transformer Fang et al. (2019). As discussed in the main paper, the SMT model consists of a scene memory for storing the visual features $\{x_i\}_{i=0}^t$ and agent poses $\{p_i\}_{i=0}^t$ seen during an episode. The environment encoder uses self-attention on the scene memory to generate a richer set of environment embeddings $\{e_i\}_{i=1}^t$. The policy decoder attends to the environment embeddings using the inputs $o_{t+1}$, which consist of

the visual feature $x$, and agent pose $p$ at time $t + 1$. The outputs of the policy decoder are used to sample an action $a_{t+1}$ and estimate the value $v_{t+1}$. Next, we discuss the details of the individual components.

**SCENE MEMORY**    It stores the visual features derived from the input images and the agent poses at each time-step. Motivated by the ideas from Sax et al. (2020), we use mid-level features derived from various pre-trained CNNs for each input modality. In this work, we consider two input modalities: RGB, and depth. For RGB inputs, we extract features from the pre-trained models in the max-coverage set proposed in Sax et al. (2020). These include surface normals, keypoints, semantic segmentation, and 2.5D segmentation. For depth inputs, we extract features from pre-trained models that predict surface normals and keypoints from depth (Zamir et al., 2020). For simplicity, we assume that the ground-truth pose is available to the agent in the form of $(x_t, y_t, z_t, \theta_t)$ at each time-step, where $\theta_t$ is the agent heading. While this can be relaxed by following ideas from state-of-the-art approaches to Neural SLAM (Chaplot et al., 2020b; Ramakrishnan et al., 2020a), we reserve this for future work as it is orthogonal to our primary contributions.

**ATTENTION MECHANISM**    Following the notations from Vaswani et al. (2017), we define the attention mechanism used in the environment encoder and policy decoder. Given two inputs $X \in \mathbb{R}^{n_1 \times d_x}$ and $Y \in \mathbb{R}^{n_2 \times d_y}$, the attention mechanism attends to $Y$ using $X$ as follows:

$$\text{Attn}(X, Y) = \text{softmax}\left(\frac{Q_X K_Y^T}{\sqrt{d_k}}\right) V_Y \tag{7}$$

where $Q_X \in \mathbb{R}^{n_1 \times d_k}, K_Y \in \mathbb{R}^{n_2 \times d_k}, V_Y \in \mathbb{R}^{n_2 \times d_v}$ are the queries, keys, and values computed from $X$ and $Y$ as follows: $Q_X = XW^q$, $K_Y = YW^k$, and $V_Y = YW^v$. $W^q, W^k, W^v$ are learned weight matrices. The multi-headed version of Attn generates multiple sets of queries, keys, and values to obtain the attended context $C \in \mathbb{R}^{n_1 \times d_v}$.

$$\text{MHAttn}(X, Y) = \text{FC}([\text{Attn}^h(X, Y)]_{h=1}^{H}). \tag{8}$$

We use the transformer implementation from PyTorch (Paszke et al., 2019). Here, the multi-headed attention block builds on top of MHAttn by using residual connections, LayerNorm (LN) and fully connected (FC) layers to further encode the inputs.

$$\text{MHAttnBlock}(X, Y) = \text{LN}(\text{MLP}(H) + H) \tag{9}$$

where $H = \text{LN}(\text{MHAttn}(X, Y) + X)$, and MLP has 2 FC layers with ReLU activations. The environment encoder performs self-attention between the features stored in the scene memory to obtain the environment encoding $E$.

$$E = \text{EnvironmentEncoder}(M) = \text{MHAttnBlock}(M, M). \tag{10}$$

The policy decoder attends to the environment encodings $E$ using the current observation $x_t, p_t$.

$$\text{PolicyDecoder}([x_t, p_t], E) = \text{MHAttnBlock}(\text{FC}([x_t, p_t]), E) \tag{11}$$

We transform the pose vectors $\{p_i\}_{i=1}^{n}$ from the scene memory relative to the current agent pose $p_t$ as this allows the agent to maintain an egocentric view of past inputs Fang et al. (2019).

## D  HYPERPARAMETERS

We detail the list of hyperparameter choices for different tasks and models in Tab. 5. For the random masking baseline in Tab. 1, we tried masking out 10, 20 and 50 frames and picked 10 frames based on the zone prediction performance. For SMT (Video), we choose 25 frames as inputs and 15 frames as output based on Dense Predictive Coding Han et al. (2019).

| RL Optimization | |
| --- | --- |
| Optimizer | Adam |
| Learning rate | 0.00025 - 0.001 |
| # parallel actors | 64 |
| PPO mini-batches | 2 |
| PPO epochs | 2 |
| PPO clip param | 0.2 |
| Value loss coefficient | 0.5 |
| Entropy coefficient | 0.01 |
| Advantage estimation | GAE |
| Normalized advantage? | Yes |
| Training episode length | 1000 |
| GRU history length | 128 |
| # training steps (in millions) | 15 |
| RNN hyperparameters | |
| Hidden size | 128 |
| RNN type | LSTM |
| Num recurrent layers | 2 |
| SMT hyperparameters | |
| Hidden size | 128 |
| Scene memory length | 500 |
| # attention heads | 8 |
| # encoder layers | 1 |
| # decoder layers | 1 |
| Occupancy memory hyperparameters | |
| Action space range | $48m \times 48m$ |
| # global action sampling interval | 25 |
| Reward scaling factors for different tasks | |
| Task | Reward scale |
| Area coverage | 0.3 |
| Flee | 1.0 |
| Object coverage | 1.0 |
| Self-supervised learning optimization | |
| Optimizer | Adam |
| Learning rate | 0.0001 |
| Video batch size | 20 |
| Temperature ($\tau$) | 0.1 |

Table 5: Hyperparameters for training our RL and self-supervised learning models.

## E  DOWNSTREAM TASK PERFORMANCE VS. EPISODE TIME

We show the downstream task performance as a function of time in Fig. 6. We evaluate each model with 3 different random seeds and report the mean and the 95% confidence interval in the plots.

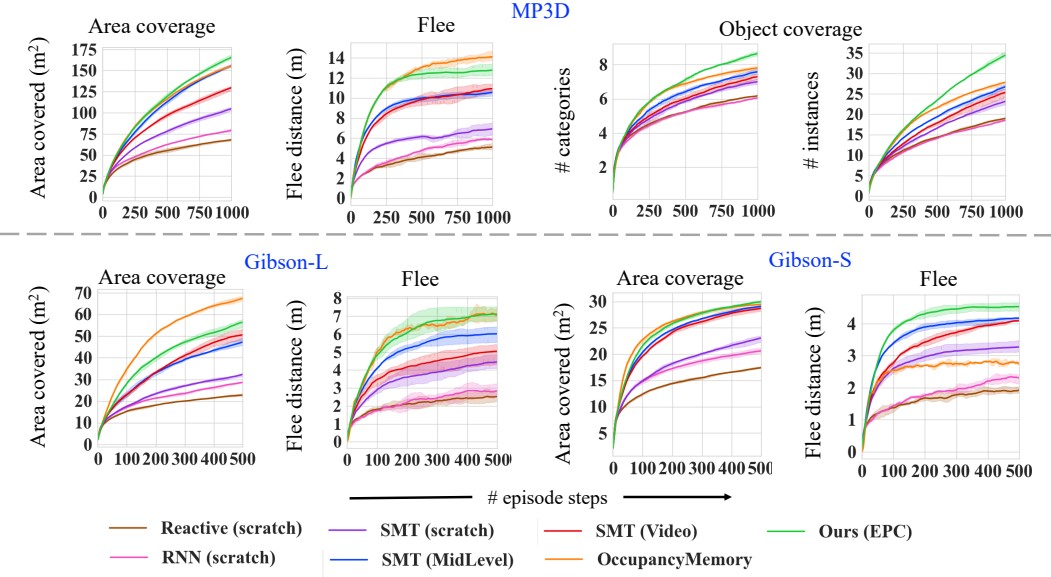

Figure 6: We highlight the downstream task performance as a function of episode time on both Matterport3D and Gibson.

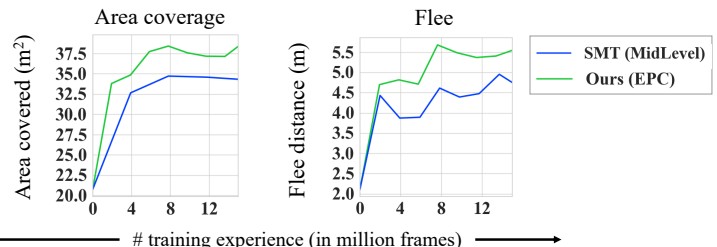

Figure 7: **Sample efficiency** on Gibson val split. Our environment-level pre-training leads to 2-4× training sample efficiency when compared to SoTA image-level pre-training.

## F    SAMPLE EFFICIENCY CURVES ON GIBSON

We plot the Gibson validation performance as a function of training experience in Fig. 7. EPC achieves better sample efficiency through environment-level pre-training when compared to the image-level pre-training baseline SMT (MidLevel).

## G    COMPLETE ANALYSIS OF NOISE ROBUSTNESS IN DOWNSTREAM TASKS

In Tab. 4 from the main paper, we compared the noise robustness of top three approaches on MP3D. Here, we present the complete set of results for all methods on Gibson and MP3D in Tab. 6.

## H    EXPLORING SPATIAL CONTEXT FOR SELF-SUPERVISION IN EPC

Originally, our EPC proposal previews context from large parts of the video (spanning several zones) to fill in the content for a missing zone (termed "EPC-global"). However, we can also leverage local spatial context spanning a limited set of frames. The SMT (Video) baseline exploits local temporal context which spanned 25 + 15 frames to derive self-supervision. Now, we consider a local variant of EPC that performs spatial reasoning within a similar context, i.e., it takes 25 frames + their poses

| | **Matterport3D** | | | | | | | | |
| | Area coverage (m²) | | | Flee (m) | | | Object cov. (#cat.) | | |
| Method | NF | N-D | N-D,P | NF | N-D | N-D,P | NF | N-D | N-D,P |
|---|---|---|---|---|---|---|---|---|---|
| Reactive (scratch) | $68.0 \pm 1.3$ | $65.8 \pm 1.4$ | $65.7 \pm 1.5$ | $5.1 \pm 0.3$ | $5.3 \pm 0.2$ | $5.3 \pm 0.2$ | $6.2 \pm 0.0$ | $6.0 \pm 0.0$ | $6.0 \pm 0.0$ |
| RNN (scratch) | $79.0 \pm 2.0$ | $74.0 \pm 0.8$ | $73.4 \pm 1.3$ | $5.9 \pm 0.0$ | $5.9 \pm 0.3$ | $6.0 \pm 0.2$ | $6.0 \pm 0.0$ | $5.9 \pm 0.0$ | $5.9 \pm 0.0$ |
| SMT (scratch) | $104.8 \pm 2.2$ | $101.6 \pm 0.9$ | $99.2 \pm 2.9$ | $6.9 \pm 0.6$ | $6.6 \pm 0.2$ | $7.4 \pm 0.2$ | $7.0 \pm 0.2$ | $6.8 \pm 0.1$ | $6.7 \pm 0.1$ |
| SMT (MidLevel) | $155.7 \pm 2.0$ | $145.1 \pm 2.3$ | $134.2 \pm 1.8$ | $10.6 \pm 0.3$ | $10.6 \pm 0.6$ | $10.8 \pm 0.4$ | $7.6 \pm 0.2$ | $7.3 \pm 0.1$ | $7.3 \pm 0.2$ |
| SMT (Video) | $129.7 \pm 2.8$ | $118.8 \pm 1.6$ | $118.4 \pm 1.9$ | $10.9 \pm 0.4$ | $9.9 \pm 0.0$ | $9.2 \pm 0.4$ | $7.3 \pm 0.2$ | $7.2 \pm 0.0$ | $7.2 \pm 0.1$ |
| OccupancyMemory | $155.6 \pm 1.4$ | $86.6 \pm 2.2$ | $85.2 \pm 2.4$ | $\mathbf{14.1} \pm 0.6$ | $10.9 \pm 0.2$ | $10.2 \pm 0.3$ | $7.8 \pm 0.1$ | $5.8 \pm 0.0$ | $5.8 \pm 0.0$ |
| EPC | $\mathbf{165.6} \pm 2.8$ | $\mathbf{148.4} \pm 1.4$ | $\mathbf{152.3} \pm 2.2$ | $12.8 \pm 0.6$ | $\mathbf{12.2} \pm 0.1$ | $\mathbf{11.4} \pm 0.2$ | $\mathbf{8.6} \pm 0.1$ | $\mathbf{8.4} \pm 0.2$ | $\mathbf{8.4} \pm 0.2$ |

| | **Gibson-S** | | | | | | | | |
| | Area coverage (m²) | | | Flee (m) | | | Object cov. (#cat.) | | |
| Method | NF | N-D | N-D,P | NF | N-D | N-D,P | NF | N-D | N-D,P |
|---|---|---|---|---|---|---|---|---|---|
| Reactive (scratch) | $17.4 \pm 0.2$ | $17.8 \pm 0.4$ | $17.8 \pm 0.4$ | $1.9 \pm 0.1$ | $1.8 \pm 0.1$ | $1.8 \pm 0.1$ | - | - | - |
| RNN (scratch) | $20.6 \pm 0.4$ | $21.5 \pm 0.3$ | $21.6 \pm 0.2$ | $2.3 \pm 0.2$ | $2.2 \pm 0.2$ | $2.2 \pm 0.2$ | - | - | - |
| SMT (scratch) | $23.0 \pm 0.7$ | $23.5 \pm 0.4$ | $23.4 \pm 0.4$ | $3.3 \pm 0.2$ | $3.3 \pm 0.0$ | $2.8 \pm 0.1$ | - | - | - |
| SMT (MidLevel) | $29.1 \pm 0.1$ | $30.8 \pm 0.4$ | $30.8 \pm 0.6$ | $4.2 \pm 0.0$ | $4.1 \pm 0.0$ | $3.4 \pm 0.0$ | - | - | - |
| SMT (Video) | $28.7 \pm 0.5$ | $30.0 \pm 0.4$ | $30.2 \pm 0.6$ | $4.1 \pm 0.0$ | $3.6 \pm 0.1$ | $3.4 \pm 0.2$ | - | - | - |
| OccupancyMemory | $29.4 \pm 0.0$ | $30.8 \pm 0.3$ | $30.6 \pm 0.2$ | $2.8 \pm 0.0$ | $3.1 \pm 0.0$ | $3.0 \pm 0.2$ | - | - | - |
| EPC | $\mathbf{29.9} \pm 0.3$ | $\mathbf{31.8} \pm 0.1$ | $\mathbf{31.6} \pm 0.2$ | $\mathbf{4.5} \pm 0.1$ | $\mathbf{4.2} \pm 0.2$ | $\mathbf{4.2} \pm 0.2$ | - | - | - |

| | **Gibson-L** | | | | | | | | |
| | Area coverage (m²) | | | Flee (m) | | | Object cov. (#cat.) | | |
| Method | NF | N-D | N-D,P | NF | N-D | N-D,P | NF | N-D | N-D,P |
|---|---|---|---|---|---|---|---|---|---|
| Reactive (scratch) | $22.8 \pm 0.6$ | $22.4 \pm 0.2$ | $22.4 \pm 0.2$ | $2.5 \pm 0.3$ | $2.6 \pm 0.4$ | $2.6 \pm 0.4$ | - | - | - |
| RNN (scratch) | $28.6 \pm 0.3$ | $27.9 \pm 2.4$ | $28.2 \pm 2.5$ | $2.8 \pm 0.4$ | $2.7 \pm 0.4$ | $2.8 \pm 0.4$ | - | - | - |
| SMT (scratch) | $32.3 \pm 0.8$ | $33.4 \pm 1.2$ | $32.6 \pm 1.9$ | $4.4 \pm 0.4$ | $4.6 \pm 0.2$ | $4.4 \pm 0.1$ | - | - | - |
| SMT (MidLevel) | $47.2 \pm 1.6$ | $49.2 \pm 0.4$ | $46.8 \pm 2.8$ | $6.0 \pm 0.4$ | $5.4 \pm 0.4$ | $5.1 \pm 0.6$ | - | - | - |
| SMT (Video) | $50.6 \pm 2.6$ | $50.4 \pm 1.4$ | $47.5 \pm 1.7$ | $5.0 \pm 0.6$ | $5.2 \pm 0.4$ | $4.4 \pm 0.4$ | - | - | - |
| OccupancyMemory | $\mathbf{67.4} \pm 0.9$ | $\mathbf{56.8} \pm 0.8$ | $\mathbf{56.9} \pm 0.8$ | $7.0 \pm 0.4$ | $6.9 \pm 0.4$ | $6.9 \pm 0.3$ | - | - | - |
| EPC | $56.4 \pm 2.1$ | $55.8 \pm 0.6$ | $55.0 \pm 0.8$ | $\mathbf{7.1} \pm 0.4$ | $\mathbf{7.0} \pm 0.6$ | $\mathbf{7.4} \pm 0.6$ | - | - | - |

Table 6: Comparing robustness to sensor noise on downstream tasks in Gibson and Matterport3D. Note: NF denotes noise free sensing, N-D denotes noisy depth (and noise-free pose), and N-D,P denotes noisy depth and pose.

| | Area coverage (m²) | | | Flee (m) | | | Object coverage (#obj) | |
| Method | Gibson-S | Gibson-L | MP3D | Gibson-S | Gibson-L | MP3D | MP3D-cat. | MP3D-inst. |
|---|---|---|---|---|---|---|---|---|
| SMT (Video) | $28.7 \pm 0.5$ | $50.6 \pm 2.6$ | $129.7 \pm 2.8$ | $4.1 \pm 0.0$ | $5.0 \pm 0.6$ | $10.9 \pm 0.4$ | $7.3 \pm 0.2$ | $25.4 \pm 1.0$ |
| EPC-local | $\mathbf{30.9} \pm 0.2$ | $\mathbf{58.0} \pm 0.4$ | $\mathbf{165.0} \pm 2.0$ | $\mathbf{4.7} \pm 0.1$ | $\mathbf{7.8} \pm 0.2$ | $\mathbf{12.9} \pm 0.9$ | $8.2 \pm 0.0$ | $32.6 \pm 0.6$ |
| EPC-global | $30.0 \pm 0.3$ | $56.4 \pm 2.1$ | $\mathbf{165.6} \pm 2.8$ | $4.5 \pm 0.1$ | $7.1 \pm 0.4$ | $12.8 \pm 0.6$ | $\mathbf{8.6} \pm 0.1$ | $\mathbf{34.5} \pm 0.8$ |
| EPC-local + S. | $31.0 \pm 0.2$ | $61.2 \pm 1.6$ | $167.9 \pm 2.1$ | $4.1 \pm 0.2$ | $7.5 \pm 0.4$ | $12.6 \pm 0.2$ | $8.4 \pm 0.1$ | $32.9 \pm 0.4$ |
| EPC-global + S | $\mathbf{31.5} \pm 0.1$ | $\mathbf{62.2} \pm 1.0$ | $\mathbf{172.4} \pm 0.6$ | $\mathbf{4.4} \pm 0.0$ | $\mathbf{8.0} \pm 0.4$ | $12.6 \pm 0.2$ | $\mathbf{8.9} \pm 0.1$ | $\mathbf{36.4} \pm 1.0$ |
| EPC aug. + S | $\mathbf{31.8} \pm 0.2$ | $\mathbf{68.0} \pm 1.6$ | $\mathbf{179.8} \pm 1.4$ | $4.4 \pm 0.2$ | $7.4 \pm 0.2$ | $\mathbf{13.2} \pm 0.3$ | $\mathbf{9.0} \pm 0.0$ | $\mathbf{37.3} \pm 0.0$ |

Table 7: Exploring spatial context for self-supervision in EPC

as inputs, and predicts the average feature for the next 15 frames conditioned on the pose. We term this as "EPC-local". We compare the two EPC variants with SMT (Video) in Tab. 7.

As expected, both EPC variants outperform SMT (Video) by a large margin, validating the main hypothesis in EPC that spatial reasoning during self-supervision is critical. However, at a first glance, it appears that EPC-global only offers limited advantage over EPC-local. Our analysis reveals that EPC-global is bottlenecked by the averaging of zone features during self-supervison (Eqn. 1). Each zone typically contains anywhere from 5 - 305 frames (mean of 48 frames), and averaging them reduces the self-supervision available per video. To test this hypothesis, we make a simple change where we replace feature averaging with sampling, i.e., we sample a random frame from the masked zone as the prediction target in Eqn. 1. The performance of this new "sampling-based" zone representation is shown in Tab. 7 (denoted as "+ S"). As expected, removing the feature averaging improves both the EPC variants. We see larger improvements in EPC-global since a lot more frames were averaged over in this case (i.e., more information lost). As we noted, these two variants capture two types of contextual cues: local and global which could be complementary. To test this, we now combine the two losses during SSL training (EPC aug. + S). This model generally outperforms the individual methods, confirming our intuition that we can derive complementary cues from local and global context.

