# OpenReview forum: "Environment Predictive Coding for Embodied Agents"
_ICLR.cc/2021/Conference — Reject_

### Official Review · AnonReviewer3 · 2020-10-23
**Good paper, interesting idea, needs more baselines**

**Rating:** 5
**Confidence:** 4

**Review:**

**Summary**

The paper proposes a self-supervised approach for learning environment-level representations for embodied agents. The idea is that agents collect images and their corresponding poses during a walk-through phase. The images are clustered into multiple "zones". The zones are divided into seen and unseen zones. Using contrastive learning, the model is trained to distinguish the features of an unseen zone from the rest of the zones. The paper shows this approach improves performance over a number of baselines for Area Coverage, Flee, and Object Coverage tasks.

**Strengths**

- The idea of environment-level representation learning is interesting.

- The paper shows improvements over a number of strong baselines for three tasks.

- The experiments show sample efficiency compared to image-level representation learning.

**Weaknesses**

- It is not clear if it is the zone prediction task that provides the improvements. While the baselines are very helpful to better understand the model, a few important baselines are missing. The following baselines clarify if the improvements can be attributed to zone prediction or not:

    (a) Train with random frames from the training environments in a contrastive learning-based framework such as MoCo (He et al., CVPR 2020) and use that as the environment encoder.

    (b) Augment SMT (Video) baseline with pose.

- There are a number of statements that are not correct and should be removed. For example,

    (a) In Section 3-zone generation, it is mentioned that "Note that the zones segment video trajectories, not floorplan maps, since we do not assume access to the full 3D environment", while the approach uses all features of a full 3D environment: computing the 3D intersection of point clouds, knowing which poses are valid and which are not, etc.

   (b) It is mentioned that "Our model competes strongly with a task-specific representation model on the tasks that the latter was designed for, while outperforming it significantly on other tasks". This model has been trained using area coverage data, which is the same or very similar to the proposed end tasks.

**Score justification**

Overall, I am positive about this paper. The main issue is that it is not clear what provides performance improvements. This should be clarified in the rebuttal.

**After rebuttal**

The rebuttal does not address my concerns so I lower my rating due to the following reason:
It was not clear where the improvement comes from so I asked for an image-level baseline, which is trained using contrastive methods. The rebuttal does not provide that. It is mentioned that adding the image-level baseline to the proposed approach even improves the results without providing any evidence. My concern was that an image-based method trained in a similar way might provide the same results. I cannot really judge if the proposed method is effective or not due to lack of this baseline. Several previous embodied representation learning works are outperformed by simple image-level baselines.

---

> ### Author Response · Authors · 2020-11-20
> **Response to Reviewer #3**
>
> We thank the reviewer for the helpful feedback.
>
> Q1. Is zone prediction providing the improvements? While the baselines are very helpful, a few are missing.
>
> (a) MoCo (He et al., CVPR 2020) was proposed for self-supervised representation learning (SSL) of images. In contrast, we learn environment-level representations of image sequences. The difference is important: the image-level and environment-level encoders are not in competition, they can actually be complementary. While it is possible to use MoCo for learning the image encoder (which could conceivably improve our results further), we feel that the MidLevel features [1] used in "SMT (MidLevel)", "SMT (Video)" and "Ours (EPC)" are already a very strong image representation. The RGB encoders obtained from [1] were trained to perform various visual tasks on 120k fully annotated images, and achieved state-of-the-art performances on various navigation tasks. The depth encoders obtained from [2] were trained on 4M fully annotated images.
>
>
> (b) Augment SMT (Video) baseline with pose: This is an interesting suggestion. The SMT (Video) baseline encompasses methods that rely on temporal signals for SSL. If we add pose to SMT (Video), the SSL task is to predict the average feature for 15 frames conditioned on their average pose and the past 25 frames + poses. So, the model learns to reason about spatial zones, but only within the local 25+15 frames context. This can be viewed as a local variant of EPC ("EPC-local"). Our original EPC proposal previews context from large parts of the video (spanning several zones) to fill in the content for a missing zone ("EPC-global").
>
> We compare these with SMT (Video) in Tab. 7, Appendix H, and we provide the analysis here. Both EPC variants outperform SMT (Video) by a large margin, validating the main hypothesis in EPC that spatial reasoning during SSL is critical. However, at a first glance, it appears that EPC-global only offers limited advantage over EPC-local. Our analysis reveals that EPC-global is bottlenecked by the averaging of zone features during SSL (Eqn. 1). Each zone typically contains anywhere from 5 - 305 frames (mean of 48), and averaging them reduces the self-supervision available per video. To test this, we simply replace feature averaging with sampling, i.e., we sample a random frame from the masked zone as the prediction target in Eqn. 1. The performance of this new "sampling-based" zone representation is shown in Tab. 7 (denoted as "+ S"). As expected, removing the feature averaging improves both the EPC variants. We see larger improvements in EPC-global since a lot more frames were averaged over in this case (more information lost). As we noted, these two variants capture two types of contextual cues: local and global which could be complementary. When we combine the two losses during SSL training (EPC aug. + S), it generally outperforms the individual methods, confirming our intuition. We have update this in Appendix H. We will incorporate it into the main text upon acceptance.
>
>
> Q2.a The approach uses all features of a full 3D environment: computing the 3D intersection of point clouds, knowing which poses are valid and which are not, etc.
>
> We think there may be a misunderstanding here. For the self-supervised learning algorithm, we **do not** assume access to 3D meshes, only short videos which contain 500 frames of RGB-D + pose each. Note that this is fairly limited information as we only have access to those poses which were visited by the data collection agent, which is outside the control of the SSL algorithm. Our new ablations in Sec. 4.3 show that EPC's performance is fairly robust to noise in this data, and that the agent gathering the data need not be intelligent (even simple heuristic policies are enough).
>
>
> Q2.b This model has been trained using area coverage data, which is the same or very similar to the proposed end tasks.
>
> As we note in Sec. 4 (experimental setup and tasks), the three tasks require different types of geometric and semantic reasoning and are not similar. Area coverage requires visiting large open spaces. Flee requires finding long corridors and travelling to distant corners. Object coverage requires semantic reasoning to discover newer object categories. For completeness, we also perform ablation studies (updated in Sec. 4.3) which indicate that our performance remains stable even when the data is collected by a simple heuristic agent that is not learned. Therefore, our approach does not rely on the similarity between tasks performed in the video walkthroughs and the downstream tasks.
>
>
> [1] Sax, et al. "Learning to Navigate Using Mid-Level Visual Priors." Conference on Robot Learning. 2020.
> [2] Zamir, et al. "Robust Learning Through Cross-Task Consistency." Proceedings of the IEEE/CVF Conference on Computer Vision and Pattern Recognition. 2020.

---

### Official Review · AnonReviewer4 · 2020-10-28
**Self-supervised approach is useful. However, how it helps embodied agents is not clear.**

**Rating:** 4
**Confidence:** 3

**Review:**

The paper studies the embodied agent problem. The paper provides a self-supervised approach to learn environment-level representations for embodied agents. For the self-supervised module, the paper considers a series of images. The paper is well written and clear. The problem studied in the paper is an important problem. The approach of adding signals is reasonable. The experimental results show nice improvement compared to existing approaches.

The motivation of using self-supervised learning seems to be clear but not very strong for this problem. Because, taking searching for a refrigerator for example, it is not clear how these self-supervised tasks enhance the reasoning ability. It is a bit of brute-force solution to have this auxiliary task. It is better to have ablation study for leave-one-out testing. In addition, if there are more auxiliary tasks, will the performance continue to be improved? How these tasks help the embedded agent is not clear. It would be more interesting to have analysis on that.

The training part is not very clear. What is the policy for navigation tasks and what is the setup?  There is no study about the effect of EPC designs. It is better to have more ablation study for EPC. The paper shows it helps improve the performance. However, how can it improve the RL training is not clear. In other words, it is better to provide more details at least experimental results about the connection between the designs of EPC and navigation tasks.

---

> ### Author Response · Authors · 2020-11-15
> **Certain points in R4's feedback are not clear to us. Can R4 clarify these points?**
>
> We thank R4 for the helpful feedback. We are planning to provide a complete response to reviewer R4 and the others, however certain points are not clear to us and we hope R4 can clarify them.
> Specifically, we are uncertain about the comments regarding the "brute-force solution", the suggestion of "leave-one-out testing" and the indication of the existence of multiple auxiliary tasks. It is possible there is a misunderstanding about what our method does.
>
> Please note that we are proposing to learn and evaluate our novel environment predictive coding (EPC) in a transfer learning setup. In this setup, we have only one self-supervised learning task that is used during pre-training, i.e., our masked zone prediction task that learns environment-level representations from a collection of egocentric walkthroughs performed by other agents (see Eqn. 3 for the self-supervised loss).
> We stress that the zone prediction task is *not* used as an auxiliary loss during RL training on the downstream tasks (i.e., area coverage, flee, and object coverage).
> For each of the downstream tasks, we independently train an RL model using its own task-specific reward (see Page 6, 3rd paragraph).  In the case of our EPC-based models, we transfer the parameters of the single model learned in the zone prediction task and fine tune it on the downstream one, as is the case with the standard transfer learning setup (Sec. 3.2, last paragraph). We don't conduct any joint training with the downstream task.
>
>  We hope that R4 can clarify the following points to us so we may best respond:
> - "It is a bit of a brute-force solution to have this auxiliary task" --- What aspect of the masked zone prediction task does R4 consider to be a brute-force solution?
> - "It is better to have ablation study for leave-one-out testing" --- Can R4 elaborate how the leave-one-out testing can be conducted in our setup?  What would be the entities to leave out in turn?
> - "If there are more auxiliary tasks, will the performance continue to be improved?" --- We are uncertain what R4 is referring to with "more auxiliary tasks"?  As described above, we have only one SSL task.
>
> We look forward to the clarification of the previous points. This would enable us to provide a complete response which addresses the above, as well as other concerns that were raised about the motivation, task setup, and ablation studies of EPC design.

---

> > ### Author Response · Authors · 2020-11-20
> > **Response to Reviewer #4**
> >
> > Pending the requested clarifications in our earlier post on 15th Nov, 2020 for some of the points raised by R4, we address the remaining ones below.
> >
> > Q1. The motivation of using self-supervised learning seems to be clear, but not very strong for this problem. Because, taking searching for a refrigerator for example, it is not clear how these self-supervised tasks enhance the reasoning ability.
> >
> > To be clear, we transfer to multiple downstream tasks, but we have only one self-supervised task in EPC: masked zone prediction (see Sec. 3.1). Please see our earlier response on the transfer setup. The key idea in this task is to learn environment-level representations of observed content in the scene, so that they are predictive of missing content (the masked zones). By performing such predictions, the agent learns the natural priors of 3D environments, which it can leverage to perform new tasks (Sec. 3.1, 1st paragraph). The refrigerator example is a good way to understand this (Page 2, 1st paragraph). Given observations from the living room and dining room, and a camera pose from the kitchen (which was not observed), the agent learns to infer the features of a refrigerator that may be present in the kitchen zone. By learning to reason about the geometry and semantics of 3D environments during self-supervised learning, the agent is able to learn faster and perform better on downstream navigation tasks.
> >
> > Q2. The training part is not very clear. What is the policy for navigation tasks and what is the setup?
> >
> > We use a Scene Memory Transformer [1] (SMT) as the navigation policy (see Sec. 3.2 and Fig. 3).
> > We pre-train the environment encoder in the SMT through self-supervision on 4000 videos generated by an agent navigating in Gibson training environments and finetune it on 3 downstream tasks of area coverage, flee, and object coverage on 72 Matterport3D training environments (see Sec. 4, experimental setup and tasks). Note that we learn one policy per downstream task, and we do not use self-supervised losses during the RL training. We evaluate the RL models on Gibson validation environments and Matterport3D test environments (Sec. 4, experimental setup). Please see Sec. 3.2 for more details on the transfer and RL training, and Appendix D for the hyperparameters.
> >
> >
> > Q3. There is no study about the effect of EPC designs. It is better to have more ablation study of EPC.
> >
> > In Sec. 4.3, we have added an ablation study that evaluates the sensitivity of self-supervised learning in EPC to (1) noisy depth, (2) noisy pose, and (3) the exploration strategy for data collection. Our results indicate that the performance on the downstream tasks is not impacted much when noisy sensors are used and data is gathered from a simple non-learned heuristic policy. R4 has not mentioned any specific ablations; we would be happy to consider a specific suggestion.
> >
> > As indicated in our previous response to R4, we are not clear about certain points in R4's feedback about "brute-force solution", "leave-one-out testing" and the existence of multiple auxiliary tasks. If R4 could clarify these points, it would help us address them more concretely.
> >
> >
> > [1] Fang et al. "Scene memory transformer for embodied agents in long-horizon tasks." Proceedings of the IEEE Conference on Computer Vision and Pattern Recognition. 2019.

---

### Official Review · AnonReviewer2 · 2020-10-28
**Official Blind Review R2**

**Rating:** 6
**Confidence:** 4

**Review:**

#############################################################################################
Summary:

This paper proposes Environment Predictive Coding which leverages predictable information from video trajectories of egocentric movements to learn the environment features in a self-supervised manner. They then show the learned environment feature encoder can be useful for downstream navigation tasks.

#############################################################################################

Pros:
1.	Learning the predictive features from the environment is an interesting idea for using self-supervised signals from the video trajectories.
2.	Different components and design decisions are well-motivated, such as noise-contrastive estimation, STM, etc.

#############################################################################################

Cons:
1.	The approach assumes training data of video trajectories from other agents in the environment. However, it is not clear to me how this dataset was generated. Could you please provide more details on how this is generated? What policy do you use? What is the task or reward these agents use?
2.	Following on the question above, another concern that I have is that the training data requires a strong exploratory policy which will be hard to achieve with a random policy. This layer of complexity is ignored in the paper. Moreover, the authors state that their approach is much less constrained than previous methods that generate data with task-specific algorithms. If this paper requires a similar level of computation and strong task similarity with the test task, then this argument should be weakened. Or at least, an analysis should be provided to detail the difference.

#############################################################################################

Recommendation and Explanation:

I recommend acceptance since the proposed self-supervised task is interesting and could be useful for the embodiment navigation community. However, I strongly recommend the authors to resolve my questions above.

---

> ### Author Response · Authors · 2020-11-20
> **Response to Reviewer #2**
>
> We thank the reviewer for the helpful feedback.
>
> Q1. It is not clear how the video dataset was generated.
>
> To generate the video data, we use an SMT (scratch) agent trained with an area-coverage reward in MP3D training environments. We then deploy it on previously unseen Gibson train environments to collect the videos only. These details are available in Sec. 4, experimental setup, and we have clarified them further to focus on the salient points.
>
>
> Q2. The training data requires a strong exploratory policy which will be hard to achieve with a random policy. Is a strong task similarity needed to gather data?
>
> No, a strong task similarity is not needed. For simplicity, we used a standard off-the-shelf area-coverage policy to generate the data. However, we appreciate the reviewer's concern.  To study the impact of that policy, we ran experiments where the "strong" area coverage policy is replaced with a much more naive, non-learned heuristic. We have added this analysis in Sec. 4.3, where we find that it is not necessary to have a strong exploration policy or strong task similarity. EPC trained on data collected by the simple heuristic performs similar to the original EPC trained on data collected by the strong exploration policy.

---

### Official Review · AnonReviewer1 · 2020-10-29
**Interesting method, some issues with evaluation**

**Rating:** 6
**Confidence:** 4

**Review:**

This paper presents a self-supervised method to learn environment-level representations for embodied agents. The representations are learned using a zone prediction task. This task involves predicting features of the masked portions of input trajectories based on unmasked portions. Results on downstream navigation tasks show an improvement over a range of baselines.

Strengths:
- The method is novel and original to the best of my knowledge. Although it is demonstrated to be useful for exploration tasks, it can potentially be useful for different types of embodied tasks.
- The paper is written well and easy to follow in most places.
- The authors have done an incredible job of reviewing related work comprehensively and discussing the proposed approach in the context of prior work.
- The authors also compare the proposed method with a wide range of useful baselines.

Weaknesses:
- I believe the main weakness of the method is the requirement of depth and pose as input during the self-supervised pretraining part. This severely limits the applicability of the method. As mentioned in the paper, the motivation is to use `in-the-wild consumer videos’ that are readily available for self-supervised pre-training. However, consumer videos typically do not have pose or depth. The zone generation process relies heavily on accurate depth and pose input. I believe it is not easy to adapt the method when to videos where depth and pose are not available or are noisy.
- The downstream navigation tasks also assume access to the true depth and pose. This setup is also unrealistic as pose and depth from sensors are often noisy.
- The trajectories used for pre-training are generated using a model trained on an exploration task and the learned representations are tested for different exploration tasks. This is counter-intuitive, the representations should be tested on a different task such as object goal or image goal navigation. These tasks provide a better variety as compared to different variants of the exploration task even otherwise.
- Except for the SMT (video) baseline, I believe the comparison to the rest of the baselines is slightly unfair as the proposed method uses 4000 videos for pretraining which is not available to the baselines.
- OccupancyMemory baseline is based on the Active Neural SLAM model, which I believe uses RGB frames as input as compared to RGBD used in EPC. This seems to be another unfair comparison.
- The environment and dataset splits are not clear. It is mentioned that the video trajectories are generated using an agent trained on MP3D. It seems unfair to use MP3D environments for pretraining and then use the same environments for testing on downstream tasks. This is especially problematic because the proposed method leads to large gains mostly in the MP3D environment.

Questions/Comments/Suggestions:
- What is the average length of 4000 videos used of pretraining?
- Why was Occupancy Anticipation (Ramakrishnan et al. 2020b) not used as a baseline?
- Why is the performance of OccupancyMemory unusually low on Gibson-S Flee as compared to other test sets?
- It would interesting to see if the proposed EPC method can be used for pretraining with OccupancyMemory, possibly leading to even better performance.
- It would be good to see the performance of the best baseline when trained for x frames where x = 15M + number of frames in 4000 pretraining videos. This would be a good upper bound for self-supervised learning.

-- Update after author response
The authors have addressed some of my concerns, I have revised my score accordingly. I am not convinced that the method is significantly better than the baselines as it performs worse on some tasks/datasets and all the tasks in the paper are similar. Even the heuristic policy used for gathering the data is similar to the downstream tasks. The addition of other types of downstream navigation tasks such as objectnav or imagenav would make the paper much stronger. I like the noisy pose and depth experiments but there's no information on the type and amount of noise. I am assuming a zero-mean gaussian noise, which is not very realistic in my opinion. The standard deviation of the noise is not reported. I encourage the authors to add some other downstream navigation task and add realistic noise with relevant details to the camera-ready version if accepted.

---

> ### Author Response · Authors · 2020-11-20
> **Response to Reviewer #1 [1/2]**
>
> We thank the reviewer for the encouraging feedback and suggestions. We provide a 2-part response to the questions raised by R1.
>
> Q1. Requirement of depth and pose during self-supervised learning is limiting. Consumer videos typically do not contain depth or pose, or these sensors are noisy.
>
> We do **not** aim to use in-the-wild consumer video. We use data collected parallely from robots that perform their own set of tasks in various environments, similar to RobotNet [4] (see 2nd paragraph, Sec. 3.1). We assume depth and odometer sensors since they are relatively inexpensive and commonly used. To further address the reviewer's concern, we have added a new ablation in Sec. 4.3 to show our model is robust to noisy depth and pose estimates in the video data.
>
>
> Q2. The downstream navigation tasks use true depth and pose, but these sensors are often noisy.
>
> Following prior work [2,3], we inject noise in depth and pose during downstream task evaluation to measure noise robustness. In short, all learned methods are generally stable. Please see Sec. 4.4 for the full analysis.
>
>
> Q3. Counter-intuitive to pre-train on exploration trajectories and test on downstream exploration tasks. The representations should be tested on a different navigation task.
>
> Our selection of the three tasks is well motivated since they require different types of geometric and semantic reasoning,  which allows us to test the flexibility of EPC. Area coverage requires visiting large open spaces. Flee requires finding long corridors and travelling to distant corners. Object coverage requires semantic reasoning to discover newer object categories (see Sec. 4, experimental setup and tasks). We also experimentally confirm that the dependence on task similarity is negligible. In Sec. 4.3, we gather an equivalent amount of videos using a simple non-learned heuristic [2,3], and observe that self-supervised learning on this data leads to similar performance on downstream tasks. We reserve benchmarking on object and image goal navigation for future work.
>
>
>
> Q4. What is the average length of 4000 videos? Comparisons to baselines unfair as proposed method uses 4000 videos for pretraining.
>
> Each video contains ~500 frames and the dataset contains ~2M frames. Note that this is **off-policy walkthrough data** used only for representation learning, not on-policy interaction data used for policy learning. We believe this comparison is fair. However, for completeness, we re-evaluate checkpoints of "Ours (EPC)" and "SMT (video)" before 15M-2M = 13M frames to account for this additional experience. The performance at 13M frames shown below remains similar, and EPC maintains its advantages over the best baselines from Tab. 2.
>
>
> |||Area|||Flee||Objects||
> |-|:-:|:-:|:-:|:-:|:-:|:-:|:-:|:-:|
> |Method|Gib.S&nbsp;|Gib.L&nbsp;|MP3D&nbsp;|Gib.S&nbsp;|Gib.L&nbsp;|MP3D&nbsp;|MP3D-cat.&nbsp;|MP3D-inst.&nbsp;|
> SMT(Video)&nbsp;|27.9|47.3|129.1|4.1|5.0|10.9|7.1|23.8|
> EPC|30.4|57.6|157.2|4.6|8.2|13.2|8.6|34.3|
>
>
>
> Q5. It is unfair to use only RGB inputs for OccupancyMemory, but RGBD inputs for EPC.
>
> This is not what we do. OccupancyMemory uses ground-truth depth to build the map [1,2,3].  The comparison is fair (clarified in Sec.4, baselines).
>
>
> Q6. It is unfair to use MP3D for both pretraining and testing on downstream tasks.
>
> We think that there is a misunderstanding here. We train an area-coverage agent on MP3D, then gather data on Gibson train environments (para. 2, Experimental setup). This is **not pre-training**. We do not use this agent for downstream tasks, only the data it gathered for SSL. We now emphasize this in Sec. 4, experimental setup. Nevertheless, as noted in Q3, we obtain similar performance using video data collected by a non-learned simple heuristic.
>
>
> Q7. Proposed method leads to large gains mostly in MP3D.
>
> MP3D has the largest environments and differences between the models become more pronounced. Compare SMT(scratch) vs. SMT(MidLevel) vs. Ours(EPC) in Tab. 2. The pairwise gains on Gibson-S $<$ Gibson-L $<$ MP3D following the relative size ordering of environments.
>
>
> [1] Ramakrishnan et al. "Occupancy Anticipation for Efficient Exploration and Navigation." European Conference on Computer Vision. Springer, Cham, 2020.
> [2] Chen et al. "Learning Exploration Policies for Navigation." International Conference on Learning Representations. 2018.
> [3] Ramakrishnan et al. "An Exploration of Embodied Visual Exploration." arXiv preprint arXiv:2001.02192 (2020).
> [4] Dasari et al. "RoboNet: Large-scale multi-robot learning." arXiv preprint arXiv:1910.11215 (2019).

---

> > ### Author Response · Authors · 2020-11-20
> > **Response to Reviewer #1 [2/2]**
> >
> > Q8. Why is OccAnt (Ramakrishnan et al. 2020b) not used as a baseline?
> >
> > OccAnt [3] efficiently builds occupancy maps by predicting the map without seeing all of it. However, its amount of area actually seen (the area coverage) is slightly lower than the ANS(depth) baseline (which is similar to OccupancyMemory). See 2nd row of Fig. 6 in [1]. Thus prior work shows OccupancyMemory is the stronger baseline for area coverage. Note that we could potentially add the anticipation facet of OccAnt to our method or any of the baselines for faster mapping.
> >
> >
> > Q9. Why is the performance of OccupancyMemory unusually low on Gibson-S Flee?
> >
> > Our qualitative analyses indicate that the global policy overfit to large MP3D environments. It often samples far away exploration targets, relying on the local navigator to explore the spaces along the sampled direction. However, this strategy fails in the small Gibson-S environments (typically a single room). Selecting far away targets results in the local navigator oscillating in place trying to exit a single-room environment. This does not affect area coverage much because it suffices to stand in the middle of a small room and look at all sides.
> >
> >
> > Q10. Can EPC method be pretrained with OccupancyMemory, leading to better performance?
> >
> > Interesting idea. The model would have a structured map as input, possibly improving performance further.
> >
> >
> >
> > [1] Ramakrishnan et al. "Occupancy Anticipation for Efficient Exploration and Navigation." European Conference on Computer Vision. Springer, Cham, 2020.
> > [2] Chen et al. "Learning Exploration Policies for Navigation." International Conference on Learning Representations. 2018.
> > [3] Ramakrishnan et al. "An Exploration of Embodied Visual Exploration." arXiv preprint arXiv:2001.02192 (2020).
> > [4] Dasari et al. "RoboNet: Large-scale multi-robot learning." arXiv preprint arXiv:1910.11215 (2019).

---

### Author Response · Authors · 2020-11-20
**Common response to reviewers**

We thank all the reviewers for their valuable feedback and helpful suggestions. The reviewers appreciate that the idea is novel and interesting (R1, R2, R3), the writing and the related work comparisons are clear (R1, R4), the design is well-motivated (R2), that there are good improvements in performance and sample efficiency over prior work (R3, R4), and that we select strong and useful baselines (R1, R3). They have raised some questions regarding the experimental setup, the video gathering process, and ablation studies. We respond to each reviewer individually and have made minor revisions (highlighted in blue) at the appropriate locations in the paper (as indicated in the individual responses).

---

### Decision · Program_Chairs · 2021-01-07
**Final Decision**

**Decision:**

Reject

**Comment:**

The paper proposes a self-supervised method to predict the gist features of image frames during navigation of an agent supervised by depth and egomotion. The features are retargeted to train navigation policies and outperform previous methods or other pretraining schemes. The idea is related to self-supervised by feature prediction but is employed in a zone level as opposed to isolated image level. Though reasonable, in the context of the recent abundance of self-supervised prediction papers in various level of spatial visual granularity, the paper may not be of sufficient novelty to present a sizable contribution for ICLR acceptance.